# Mitochondrial Dysfunction Causes Cell Death in Patients Affected by Fragile-X-Associated Disorders

**DOI:** 10.3390/ijms25063421

**Published:** 2024-03-18

**Authors:** Martina Grandi, Chiara Galber, Cristina Gatto, Veronica Nobile, Cecilia Pucci, Ida Schaldemose Nielsen, Francesco Boldrin, Giovanni Neri, Pietro Chiurazzi, Giancarlo Solaini, Alessandra Baracca, Valentina Giorgio, Elisabetta Tabolacci

**Affiliations:** 1Department of Biomedical and Neuromotor Sciences, University of Bologna, 40126 Bologna, Italy; martina.grandi11@unibo.it (M.G.); chiara.galber@studenti.unipd.it (C.G.); cristina.gatto2@unibo.it (C.G.); ida.nielsen@studio.unibo.it (I.S.N.); giancarlo.solaini@unibo.it (G.S.); alessandra.baracca@unibo.it (A.B.); 2Institute of Neuroscience, Consiglio Nazionale delle Ricerche, 35121 Padova, Italy; 3Department of Life Sciences and Public Health, Catholic University of Sacred Heart, 00168 Rome, Italy; veronicanobile88@gmail.com (V.N.); c.pucci91@yahoo.it (C.P.); giovanni.neri43@gmail.com (G.N.); pietro.chiurazzi@unicatt.it (P.C.); elisabetta.tabolacci@unicatt.it (E.T.); 4Department of Biology, University of Padova, 35121 Padova, Italy; francesco.boldrin@unipd.it; 5UOC Medical Genetics, Fondazione Policlinico Universitario A. Gemelli IRCCS, 00168 Rome, Italy

**Keywords:** fragile-X-related disorders (FXDs), neurodegeneration, donut-shape mitochondria, apoptosis, permeability transition pore, ATP synthase

## Abstract

Mitochondria are involved in multiple aspects of neurodevelopmental processes and play a major role in the pathogenetic mechanisms leading to neuro-degenerative diseases. Fragile-X-related disorders (FXDs) are genetic conditions that occur due to the dynamic expansion of CGG repeats of the *FMR1* gene encoding for the RNA-binding protein FMRP, particularly expressed in the brain. This gene expansion can lead to premutation (PM, 56–200 CGGs), full mutation (FM, >200 CGGs), or unmethylated FM (UFM), resulting in neurodegeneration, neurodevelopmental disorders, or no apparent intellectual disability, respectively. To investigate the mitochondrial mechanisms that are involved in the FXD patients, we analyzed mitochondrial morphology and bioenergetics in fibroblasts derived from patients. Donut-shaped mitochondrial morphology and excessive synthesis of critical mitochondrial proteins were detected in FM, PM, and UFM cells. Analysis of mitochondrial oxidative phosphorylation in situ reveals lower respiration in PM fibroblasts. Importantly, mitochondrial permeability transition-dependent apoptosis is sensitized to reactive oxygen species in FM, PM, and UFM models. This study elucidated the mitochondrial mechanisms that are involved in the FXD phenotypes, and indicated altered mitochondrial function and morphology. Importantly, a sensitization to permeability transition and apoptosis was revealed in FXD cells. Overall, our data suggest that mitochondria are novel drug targets to relieve the FXD symptoms.

## 1. Introduction

Fragile-X-related disorders (FXDs) are a group of genetic conditions associated with the dynamic mutation of a trinucleotide CGG repeat in the 5′UTR of the *FMR1* gene, encoding the fragile X messenger ribonucleoprotein 1 (FMRP). Fragile X syndrome (FXS) is the most common monogenic form of inherited intellectual disability and autism [1,2] and occurs due to the presence of full mutation (FM, >200 CGGs) that becomes methylated, resulting in the absence of FMRP [3]. A distinct condition is found in individuals with premutation (PM, 56–200 CGGs) [4,5]. These individuals can develop fragile-X-associated tremor/ataxia syndrome (FXTAS) [6,7]. At the molecular level, the PM condition is characterized by decreased levels of the FMRP. Additionally, rare individuals with an apparently normal phenotype are carriers of unmethylated FM alleles (UFM) that allow FMRP production, although these are subjected to neurodegeneration [8,9]. In individuals with FXDs, most of the clinical and molecular phenotypes are linked to the absence or decreased amount of FMRP. In the brain, this RNA-binding protein associates with mRNAs and regulates multiple steps of their metabolism, mainly at the synapses [10]. FMRP functions as an mRNA-binding translation suppressor, but recent findings have enormously expanded its proposed roles to involvement in RNA-, channel-, and protein-binding leading to calcium signaling, activity-dependent critical period development, and excitation–inhibition neural circuitry balance [11,12,13].

In neurons, the energy metabolism is finely regulated, and dysfunctions of mitochondrial oxidative phosphorylation (OXPHOS) can result, on one hand, in reduced ATP levels and, on the other, in increased reactive oxygen species (ROS) production, as superoxide anion (O_2_^−•^), hydrogen peroxide (H_2_O_2_), and hydroxyl radical (OH^•^), leading to oxidative stress and Ca^2+^ deregulation. Derangements of Ca^2+^ and ROS homeostasis can induce opening of the permeability transition (PT) pore (PTP), a channel located in the inner mitochondrial membrane and dysregulated in several neurodegenerative diseases [14]. Binding to the ATP synthase of cyclophilin D (CyPD), the receptor for the PTP inhibitor cyclosporine A (CsA), and favors the Ca^2+^-dependent PTP formation [14], further promoted by thiol oxidation and membrane depolarization [15]. Pharmacological treatments causing PTP desensitization or targeting the CyPD were shown to be effective against neurodegeneration in autoimmune encephalomyelitis, superoxide dismutase 1-associated or Alzheimer’s disease models [14,16] (Figure 1).

Proteomic data show dysregulations of mitochondrial proteins in fibroblasts from UFM and PM carriers [17]. Specifically, the upregulation of the mitochondrial superoxide dismutase isoform correlated with an increased number of morphological mitochondrial abnormalities and depolarization of the mitochondrial inner membrane [17]. A decrease in basal and ATP synthesis-coupled mitochondrial respiration was found in fibroblasts derived from FXD patients [18,19,20], although this finding is controversial [21]. High levels of precursor of the mitochondrial ATP synthase β subunit were detected in the cortex of carries with FXTAS [18]. These findings are in line with the high level of the ATP synthase β and c subunits revealed in brain mitochondria of a FXS mouse model *Fmr1^-ly^* [22]. In *Fmr1^-ly^* mouse neurons, a mitochondrial membrane proton leak was shown to contribute to a metabolic shift towards glycolysis [22]. Closure of the ATP synthase proton leak channel by mild depletion of the c subunit or its pharmacological inhibition decreased lactate production and normalized synaptic maturation [22].

Considering the limited and controversial studies present in the literature on cells from PM individuals [23,24,25], it is of utmost importance to address the molecular mechanisms of mitochondrial dysfunction in cells derived from FXDs patients. This might help in the future to define novel therapeutic strategies that might potentially improve patient conditions.

## 2. Results

### 2.1. Mitochondria of FXD Patients Show Altered Morphology

FXD patient fibroblasts were analyzed to characterize their mitochondrial morphology and clarify whether the number of cristae or other membrane alterations might be a consequence of the mRNAs dysregulation. The occurrence of donut-shaped mitochondria in FXS, PM, and UFM patient fibroblasts is the most relevant abnormality revealed by transmission electron microscopy (TEM) analysis (Figure 1A–C), which is in line with previous observations by fluorescent staining of the organelles [17]. Moreover, TEM analysis shows that the number of mitochondria (Figure 1B) and the number of cristae or cristae junctions (Figure 1A,D) are not significantly different in FXS, PM, and UFM fibroblasts from that observed in controls. Nonyl acridine orange (NAO) fluorescent probe, which measures the levels of cardiolipin, was used to quantify mitochondrial membranes. FXS fibroblasts, but not PM and UFM patient cultures, show a higher level of NAO fluorescence normalized per cells or per cristae junctions (Figure 1E,F), suggesting the presence of a higher expansion of mitochondrial membranes than in controls.

### 2.2. Mitochondrial OXPHOS Complexes and Other Proteins Are Upregulated in FXDs

Mitochondrial structure and function are strictly connected. The mitochondrial OXPHOS complexes and function were investigated in cells showing mitochondrial morphology alterations. Subunits participating in the assembly of the OXPHOS complexes were detected by Western blotting (Figure 2A). FXD cell lysates show higher levels of protein complexes than controls (Figure 2A), suggesting that the OXPHOS machinery might be under the FMRP control in FXS, PM, and UFM cells. Higher levels of complex I, complex II, complex IV, and ATP synthase were observed in FXS, PM, and UFM samples than in controls (Figure 2(Aii)). In line with our findings, the accumulation of ATP synthase subunits was previously observed in a FXS mouse model [22]. The increase in OXPHOS subunits was matched by Western blotting analyses of cell lysates from a different cohort of patients carrying methylated FM, PM, or UFM alleles compared with a different healthy donor (Appendix A). However, the high amount of the OXPHOS subunits in FXS and UFM cells does not affect the mitochondrial respiration as shown by in situ measurements of the oxygen consumption rate (Figure 2B). The ATP synthesis-coupled respiration (basal) and rotenone-sensitive respiration of PM fibroblasts were significantly lower than the ones measured in control cells (Figure 2(Bii)). The lowest basal respiration in PM fibroblasts inversely correlates with the highest level of the subunits detected for complex I, II, and IV, and that of ATP synthase (Figure 2(Aii)) in the FXD genotypes, suggesting that the supernumerary complexes are not properly assembled and functional. Moreover, the measurements by TMRM fluorescence do not show significant differences in the mitochondrial membrane potential of the different cell types (Figure 2C).

Measurements of the pH acidification rate of the extracellular compartment in situ, representing the rate of glycolysis, correlated with the oxygen consumption rate in the different cell models. This suggests that the glycolytic pathway is not increased in any of the patient fibroblasts to compensate for a limitation in the mitochondrial ATP synthesis (Figure 2D).

To investigate specific mitochondrial proteins that might be involved in mitochondrial dysfunction in patients’ fibroblasts, the ATP synthase β, b, OSCP, and c subunits were independently analyzed (Figure 3A and Appendix A). Interestingly, the ATP synthase subunit c was found upregulated in FXS, PM, and UFM patients, as previously described in a FXS mouse model [22] and in line with the described proton leak in PM fibroblasts [18,19]. The ATP synthase OSCP subunit, which is the binding site for CyPD [26,27] and might be involved in the PTP modulation [28,29], is also overexpressed in FXD patient cells (Figure 3(Ai,Aii) and Appendix A). Moreover, the mitochondrial ATPase inhibitor protein IF1, the PTP modulators adenine nucleotide translocator (ANT) or CyPD, and citrate synthase were assessed (Figure 3B and Appendix A). CyPD and IF1, which represent two modulators of both the ATP synthase catalytic activity [26,30,31] and the PTP opening [26,32], are highly expressed in FXS, PM, and UFM fibroblasts (Figure 3B).

### 2.3. Cells from FXD Patients Are Sensitized to Apoptotic Stimuli

Given the overexpression of both the c and OSCP subunits of ATP synthase together with the upregulation of CyPD and the IF1 inhibitor in FXD models, the PTP opening modulation was investigated. Fibroblasts were analyzed through the Ca^2+^ retention capacity assay, which reveals the mitochondrial Ca^2+^ threshold required for PTP opening (Figure 4A). PM and UFM cells showed higher sensitivity to Ca^2+^, inducing PTP opening, while FXS mitochondrial behavior was similar to that of controls (Figure 4A). The higher Ca^2+^ threshold for PTP opening in the FXS cells, among the FXD phenotypes, correlates with higher mitochondrial membrane content, as measured by NAO fluorescent staining and normalized for cell number (Figure 1E). This might indicate a higher Ca^2+^-buffering capacity of FXS mitochondria, which is due to a higher matrix volume or to expanded mitochondrial membranes. Therefore, the Ca^2+^ levels promoting PTP opening in FXS cells may be in line with those required in PM and UFM patients if normalized to the mitochondrial membrane content. We further tested the effect of arachidonic acid, which sensitizes cells to PTP opening through channel activation by ROS and Ca^2+^ [33,34]. Apoptotic cell death was analyzed through the annexin V staining in two different cohorts of cells derived from patients and healthy donors (Figure 4B and Appendix A). Arachidonic acid treatment stimulated higher PTP-dependent apoptosis in FXS, PM, and UFM fibroblasts (Figure 4B and Appendix A). These findings indicate a higher sensitivity, upon Ca^2+^ and ROS stimuli, to PT of these cells compared to controls. The effect of an alternative treatment with Bz 423 was studied. This compound is a PTP-inducer, which was demonstrated to activate the channel by its direct binding on the OSCP subunit of ATP synthase, and to displace both CyPD and IF1 from their binding sites [26,32]. In our results, Bz 423 treatment was not efficient to induce apoptosis (Figure 4B and Appendix A), probably due to the high levels of CyPD and IF1 masking its binding site [32] in FXS, PM, and UFM fibroblasts.

This set of experiments suggests that FXD fibroblasts are sensitized to apoptotic death through mechanisms that involve PTP opening.

## 3. Discussion

This study shows that donut-shaped mitochondrial morphology and excessive synthesis of critical mitochondrial proteins are associated with the dynamic expansion of CGG repeats of the *FMR1* gene encoding for the RNA-binding protein FMRP in fragile-X-related disorders (FXDs). In FXD cells, the increased sensitivity of the permeability transition pore opening to Ca^2+^ and ROS correlates with altered mitochondrial morphology and causes higher apoptotic cell death than in healthy donors (Figure 2).

Early signs of neurodegeneration such as ubiquitin-positive inclusions were shown in UFM [8,35], as well as PM carriers who developed FXTAS [36,37]. Many neurodegenerative diseases exhibit abnormal morphology and biochemical dysfunction of mitochondria [38,39]. In FXTAS, evidence of mitochondrial dysfunctions was firstly detected in fibroblasts and brain samples of PM carriers and when present, they predisposed individuals to FXTAS [20]. Other studies supported the role of mitochondrial dysfunction in the pathogenesis of FXTAS, indicating lower mitochondrial membrane potential, and decreased basal respiration and ATP synthesis or iron accumulation in PM patients [18,19,25,40,41]. Mitochondrial dysfunction was assessed in plasma of PM carriers, displaying Warburg-like shift with increases in lactate levels and altered Krebs cycle intermediates, neurotransmitters, markers of neurodegeneration, and increases in oxidative-stress-mediated damage [42]. On the contrary, lymphoblasts of PM carriers showed a significant increase in mitochondrial respiratory activity without any sign of abnormal mitochondrial morphology [21].

In this study, donut-shaped mitochondria were confirmed and finely characterized in FXD fibroblasts through TEM analysis. As previously observed [17], the total number of mitochondria remains unmodified in all genotypes. This finding was matched by un-altered cristae number. FXS fibroblasts showed abnormal mitochondria not seen before, and expanded mitochondrial membranes. These abnormalities support a role of mitochondria in neurodevelopmental disorders such as FXS [43], and suggest that compensatory mechanisms like biogenesis might occur in this specific phenotype.

Morphological abnormalities correlated with protein dysregulation. FXD patient cells exhibited higher levels of OXPHOS and other mitochondrial proteins compared to controls. We hypothesize that protein accumulation depending on the alteration of a normal FMRP function may lead to oxidative stress. Through a comparative proteomic analysis between control, FXS, and UFM fibroblasts an upregulation of superoxide dismutase isoform 2 in UFM cells emerged, also present in PM cells [17]. In UFM and PM cells, the high superoxide dismutase isoform 2 levels correlated with morphological abnormalities (donut-shaped mitochondria). These mitochondrial phenotypes reverted after knocking-down *FMR1* transcript through siRNA, suggesting that in PM and UFM carriers, the dysregulation of protein transcripts may lead to mitochondrial dysfunction. Similar conclusions came from the work of Hukema and co-workers [44], who showed in a *Fmr1* transgenic mouse model that expanded CGG RNA expression can cause mitochondrial dysfunction [44]. Recently, these data were further supported by the finding that FMRpolyG exerted a toxic gain-of-function effect, altering mitochondrial function, bioenergetics, and initiating cell death [45,46].

In our functional studies, oxygen consumption was normal except for the basal respiration of PM fibroblasts, which was significantly lower than in control, indicating lower ATP synthesis. This result is in line with previous observations [18,19,20]. Overall, the oxygen consumption results support the hypothesis that the OXPHOS accumulation in FXD phenotypes may cause complex aggregates that are non-functional, and do not participate in the oxidative phosphorylation.

Interestingly, the subunit c of ATP synthase was found to be upregulated in all FXD fibroblasts, as previously described in a FXS mouse model [22], and according to the proton leak described in PM fibroblasts [18,19].

The binding site for CyPD [26,27] that might be involved in the PTP modulation [28,29], namely the OSCP subunit of the ATP synthase [47], is also overexpressed in FXD cells. Consistently, CyPD and IF1, two modulators of both the ATP synthase catalytic activity [26,30] and PTP opening [26,32], are significantly overexpressed in FXD fibroblasts.

The PTP opening was indeed sensitized in all the FXD cells. The Ca^2+^ threshold for the channel activation in FXS fibroblasts correlates with higher mitochondrial membrane content, as assessed by NAO fluorescent staining. This might indicate a higher Ca^2+^-buffering capacity of FXS mitochondria, which is due to a higher matrix volume or to expanded mitochondria. Therefore, the Ca^2+^ levels promoting PTP opening in FXS cells may be considered in line with those required in PM and UFM cells if normalized to the mitochondrial membrane content.

Finally, we showed that the treatment with arachidonic acid but not with Bz 423 in-creased apoptosis in FXD fibroblasts. Bz 423 is the PTP-inducer that demonstrates activating the channel by direct binding to the OSCP subunit of ATP synthase, and displaces both CyPD and IF1 from their binding sites [26,32]. This compound was not efficient in inducing cell apoptosis, probably due to the high levels of CyPD and IF1 masking its binding site in FXD fibroblasts [32]. The higher sensitivity to PTP opening in these patients might lead to neurodegenerative processes, as previously demonstrated for other neurodegenerative diseases [28,29,48,49]. Transient mitochondrial membrane depolarization or osmotic pressure have been proposed as factors promoting donut-shaped morphology [50]. Donut-shaped mitochondria formation, induced by FCCP treatment, was demonstrated to be prevented by PTP inhibitors [51], suggesting that in FXD fibroblasts, both membrane depolarization and PTP opening might promote the formation of donut-shaped mitochondria.

## 4. Materials and Methods

### 4.1. Experimental Model

Fibroblasts derived from two unaffected control males (CTR), two FXS patients, two PM, and two UFM carriers were employed in this study. Fibroblasts from FXS, UFM, and PM carriers were from skin biopsies obtained after a signed informed consent of different donors. Ethics Committee at the Catholic University of Rome approved this study (prot. N. 15152/15, approval date 1 July 2015). Cell lines were anonymously cultured in the Institute of Genomic Medicine at the Catholic University (Rome, Italy). Fibroblasts from healthy donors (CTR) were purchased by Coriell Institute. Cell cultures were grown in Dulbecco’s modified Eagle’s medium (DMEM, Thermo Fisher Scientific, Waltham, MA, USA), supplemented with fetal bovine serum (FBS, 20% *v*/*v*, Thermo Fisher Scientific), 2 mM L-glutamine, penicillin/streptomycin (1% *v*/*v*, Thermo Fisher Scientific) at 37 °C with 5% CO_2_.

### 4.2. Electron Microscopy

Patient fibroblasts were seeded at 20,000 (CTR, FXS, PM, UFM) cells/well in a 24-well plate in order to reach 70–80% of confluence. The samples were post-fixed with 1% (*v*/*v*) osmium tetroxide in 0.1 M sodium cacodylate buffer for 1 h. After three water washes, samples were dehydrated in a graded ethanol series and embedded in an epoxy resin (Sigma-Aldrich, St. Louis, MO, USA). Ultrathin sections (60–70 nm) were obtained with an Ultrotome V (LKB) ultramicrotome, counterstained with uranyl acetate and lead citrate, and viewed with a Tecnai G2 (FEI, Hillsboro, OR, USA) transmission electron microscope operating at 100 kV. Images were captured with a Veleta (Olympus Soft Imaging System, Tokyo, Japan) digital camera. Mitochondria were analyzed by counting the number of donut-shaped organelles. Cristae junctions per mitochondrion were quantified by monitoring the points of contact between the inner and the outer membranes in each mitochondrion as in [52].

### 4.3. Nonyl Acridine Orange Staining

Nonyl acridine orange (NAO) staining has been used to measure the mitochondrial membrane content in cells. The dye specifically binds cardiolipin in mitochondria. Cells were seeded at 200,000 cells/well in a 12-well tissue culture plate. NAO dye (200 nM, Merck, Darmstadt, Germany) was added to cells 24 h after seeding. Cells were stained for 30 min at 37 °C in a 5% CO_2_ humidified incubator, washed, detached with trypsin, centrifuged at 1000× *g* for 5 min, and suspended in the Hanks’ Balanced Salts (HBSS, Sigma-Aldrich) buffer. Mitochondrial mass was assessed by the Muse cell analyzer (Millipore, Burlington, MA, USA). Data acquisition and analysis were performed with MuseSoft Analysis (version 1.5) and Flowing software (version 2.5.1), respectively. A total of 5000 events were acquired for each determination.

### 4.4. Lysates, Gel Electrophoresis, Western Blotting

Cells (10 × 10^6^) were kept on ice for 20 min in 0.15 mL of a buffer containing 150 mM NaCl, 20 mM Tris, 5 mM EDTA-Tris, pH 7.4 with the addition of 1% (*v*/*v*) Triton X-100, 10% (*v*/*v*) glycerol, phosphatase, and protease inhibitors. Sample buffer (Nu-PAGE™ LDS sample buffer, Invitrogen supplemented with 12.5% *v*/*v* β-mercaptoethanol) was added to supernatants, and samples were separated by polyacrylamide gel (NuPAGE™, 12% Bis-Tris, Invitrogen, Waltham, MA, USA) electrophoresis and transferred to nitrocellulose membranes. Blocking was performed with a PBS-solution containing 5% (*w*/*v*) non-fat dry milk (AppliChem, Darmstadt, Germany). Antibodies for OXPHOS (OXPHOS Human WB Antibody Cocktail, RRID: AB_2756818), adenine nucleotide translocator isoform 3 (ANT, RRID: AB_2619664), citrate synthase (CS, RRID: AB_10678258), cyclophilin D (CyPD, RRID: AB_10864110), ATPase inhibitor (IF1, RRID: AB_10861497), and for β (RRID: AB_301438), b (RRID: AB_10901555), c (RRID: AB_2935765), and OSCP (RRID: AB_10887942) subunits were from Abcam (Cambridge, UK), and the one against GAPDH (RRID: AB_561053) was from Cell Signaling (Danvers, MA, USA). Band pixels of each replicate are normalized on band pixels of their proper loading control. Mean pixel ratios ± SEM are shown. Western blotting band intensities were analyzed using ChemiDoc MP system equipped with the ImageLab software (version 6.1, Bio-Rad, Hercules, CA, USA) or ImageJ software (version 1.52p, RRID: SCR_003070).

### 4.5. Oxygen Consumption Rate

Oxygen consumption rate (OCR) in adherent cells was measured using the XF24 Extra-cellular Flux Analyzer (Agilent Technologies, Santa Clara, CA, USA). Briefly, patient fibroblasts were seeded in XF24 cell culture microplates at 20,000 cells/well and left growing at 37 °C in a 5% CO_2_ humidified incubator for 24 h. The day after, the growth medium was replaced with the Seahorse medium (DMEM, Sigma D5030, supplemented with NaCl, glutamine, and phenol red according to the manufacturer protocol and 25 mM glucose, 10 mM sodium pyruvate), and cells were incubated at 37 °C for 30 min to allow temperature and pH equilibration. After an OCR baseline measurement, 1 µM oligomycin, 0.2–0.6 µM carbonyl cyanide-p-trifluoromethoxyphenylhydrazone (FCCP), 1 µM rotenone, and 1 µM antimycin A were sequentially added to each well. Before each experiment, a titration curve with FCCP was performed to assess the optimal FCCP concentration that maximally stimulates respiration. Acidification rate (ECAR) was monitored simultaneously to OCR when cells were in basal condition.

### 4.6. Mitochondrial Membrane Potential

Mitochondrial membrane potential was measured based on the mitochondrial accumulation of tetramethylrhodamine methyl ester (TMRM) in intact cells as in [28]. Patient fibroblasts (CTR, FXS, PM, UFM) were seeded at 200,000 cells/well in a 12-well tissue culture plate. The day after seeding, cells were incubated for 30 min at 37 °C in an FBS-free DMEM medium containing 20 nM TMRM and 1.6 μM cyclosporin H to inhibit the multidrug resistance pump. Cells were then washed, detached with trypsin, centrifuged at 1000× *g* for 5 min, and suspended in HBSS buffer. Mitochondrial membrane potential was immediately analyzed by flow cytometry using the Muse cell analyzer (Millipore, Burlington, MA, USA). Data acquisition and analysis were performed with MuseSoft Analysis and Flowing software, respectively. A total of 5000 events were acquired for each determination.

### 4.7. Calcium Retention Capacity

For the calcium retention capacity (CRC) assay, external mitochondrial Ca^2+^ was measured by Ca^2+^ Green-5N fluorescence using a Tecan Infinite^®^ 200 PRO (Tecan Trading AG, Männedorf, Switzerland) plate reader. Fibroblasts were permeabilized as described in [45] and were resuspended at the concentration of 10^7^ × mL^−1^ in a KCl-based medium (130 mM KCl, 10 mM MOPS-Tris, 10 µM EGTA) supplemented with 5 mM succinate-Tris, 1 mM Pi-Tris, and 0.5 µM Ca^2+^ Green-5N, pH 7.4 to a final volume of 0.2 mL. For all CRC measurements, sequential 5 µM CaCl_2_ pulses were added to cells.

### 4.8. Cell Death

CTR, FXS, PM, and UFM fibroblasts were seeded at 200,000 cells/well in a 12 well-tissue culture plate and then incubated with 200 μM arachidonic acid (ARA) or 100 μM benzodiazepine (Bz) 423 in a FBS-free DMEM medium at 37 °C in a 5% CO_2_ humidified incubator. After 2, 3 h (ARA) or 8 h (Bz 423) of treatment, cells were harvested with trypsin and counted. Cells incubated only with DMEM medium were harvested and counted, as control. Staurosporine (2 μM) treatment was used as positive control for cell death, and was incubated for 24 h. Cell death was assessed by the Muse cell analyzer (Millipore) using Muse Annexin V and Dead Cell kit (Luminex Flow Cytometry and Imaging, Austin, TX, USA), following the manufacturer’s instructions.

### 4.9. Quantification and Statistical Analysis

Unless stated otherwise in the figure legends, each experiment derives from at least three independent biological replicates. Data are expressed as mean  ±  SEM. *p* values indicated in the figures are calculated with GraphPad; Student’s *t* test and two-way ANOVA are applied (* is *p*  ≤  0.05, ** *p*  ≤  0.01, *** *p*  ≤  0.001). The variance between the compared groups is similar. GraphPad (version 8.0.1, RRID: SCR_002798) and Inkscape (version 1.3.0, RRID: SCR_014479) software were used to create the artwork. Schemes were created with BioRender.com (accessed on 23 February 2024).

## 5. Conclusions

In conclusion, our study shows that low levels of FMRP protein in FXD patients causes the upregulation of critical mitochondrial proteins including OXPHOS complexes, ATP synthase c, and OSCP subunits, and CyPD and the IF1 inhibitor of ATP synthase. The excessive mitochondrial protein synthesis and formation of non-functional complex aggregates may cause ROS production. In PM and UFM fibroblasts, some compensatory mechanisms occur, such as the increase in the levels of superoxide dismutase to counteract ROS formation, or that of the ATPase inhibitor IF1, which may act to preserve membrane potential and avoid mitochondrial ATP dissipation.

However, the increased sensitivity of PTP opening to Ca^2+^ and ROS in FXS, PM, and UFM patients may cause altered mitochondrial morphology and promotes higher apoptotic cell death than in controls. These results represent the basis for future investigations to better clarify the role of the described mitochondrial abnormalities both during neurodevelopment (as in FXS) and in neurodegeneration (as in PM and UFM carriers at risk of developing FXTAS). Since pharmacological trials based on promising compounds failed to show significant clinical benefits illustrating the need of novel targeted therapies [53,54,55,56], our findings suggest that novel targeted therapies for FXD patients might be focused on antioxidant defense and pharmacological inhibition of the PTP, for example the compound TR001, which inhibits the PTP independently of CyPD and prevents thiol oxidation [57], or other PTP-specific inhibitors [58,59,60].

## Data Availability

Data are contained within the article and Appendix A. The data presented in this study are available on request from the corresponding author.

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
