# Peer review of "Mitochondrial Dysfunction Causes Cell Death in Patients Affected by Fragile-X-Associated Disorders"

_ijms, 2024, doi:10.3390/ijms25063421_

Round 1

Reviewer 1 Report

Comments and Suggestions for Authors

The study conducted by the authors suggests that FXD patients with low levels of FMRP protein experience an upregulation of critical mitochondrial proteins, such as OXPHOS complexes, ATP synthase c and OSCP subunits, CyPD, and the IF1 inhibitor of ATP synthase. This excessive mitochondrial protein synthesis and formation of non-functional complex aggregates could lead to the production of ROS. Compensatory mechanisms occur in PM and UFM fibroblasts, including the increase of the levels of superoxide dismutase to counteract ROS formation or that of the ATPase inhibitor IF1, which may act to preserve membrane potential and avoid mitochondrial ATP dissipation. However, the increased sensitivity of PTP opening to Ca2+ and ROS in FXS, PM, and UFM patients may cause altered mitochondrial morphology and promote higher apoptotic cell death than controls.

These findings lay the groundwork for future investigations to understand better the role of the described mitochondrial abnormalities during neurodevelopment (as in FXS) and neurodegeneration.

The manuscript is well organized, the results and the discussion are in mutual agreement.

Minor correction:

1. In the introductory part of the manuscript, the content from line 51 to line 73 should be shown schematically for easier monitoring of the exposed material.

2. The authors could provide structural examples for reactive oxygen species.

  3. In the conclusion there is a sentence: our findings suggest that novel targeted therapies for FXD patients might be focused on antioxidant defense and pharmacological inhibition of the PTP. This should perhaps end the discussion by giving the structures of several potential drug molecules, which are most optimal according to the Authors for PTP inhibition and antioxidant defense.

Author Response

Dear Editor,
We would like to greatly thank you and the Reviewers for the opportunity to improve our work, which in our opinion has been ameliorated by the revision process.
We have addressed all the Reviewer’s points as you can find in detail in the point by point-response to Reviewers below.
Please find attached our revised manuscript and figures.
Sincerely,
Valentina Giorgio

Reviewer 1
The study conducted by the authors suggests that FXD patients with low levels of FMRP protein experience an upregulation of critical mitochondrial proteins, such as OXPHOS complexes, ATP synthase c and OSCP subunits, CyPD, and the IF1 inhibitor of ATP synthase. This excessive mitochondrial protein synthesis and formation of non-functional complex aggregates could lead to the production of ROS. Compensatory mechanisms occur in PM and UFM fibroblasts, including the increase of the levels of superoxide dismutase to counteract ROS formation or that of the ATPase inhibitor IF1, which may act to preserve membrane potential and avoid mitochondrial ATP dissipation. However, the increased sensitivity of PTP opening to Ca2+ and ROS in FXS, PM, and UFM patients may cause altered mitochondrial morphology and promote higher apoptotic cell death than controls.
These findings lay the groundwork for future investigations to understand better the role of the described mitochondrial abnormalities during neurodevelopment (as in FXS) and neurodegeneration. The manuscript is well organized, the results and the discussion are in mutual agreement.
Minor correction:
1. In the introductory part of the manuscript, the content from line 51 to line 73 should be shown schematically for easier monitoring of the exposed material.
We thank this Reviewer for the suggestion, a schematic representation of the indicated content was added in the revised version in Scheme 1, on page 2.
2. The authors could provide structural examples for reactive oxygen species.
In the revised manuscript we have added the examples for reactive oxygen species on page 2, lines 55-59 as follows:
“In neurons, the energy metabolism is finely regulated, and dysfunctions of mitochondrial oxidative phosphorylation (OXPHOS) can result on one side in reduced ATP levels and, on the other, in increased reactive oxygen species (ROS) production, as superoxide anion (O2-•), hydrogen peroxide (H2O2) and hydroxyl radical (OH•), leading to oxidative stress and Ca2+ deregulation.”
3. In the conclusion there is a sentence: our findings suggest that novel targeted therapies for FXD patients might be focused on antioxidant defense and pharmacological inhibition of the PTP. This should perhaps end the discussion by giving the structures of several potential drug molecules, which are most optimal according to the Authors for PTP inhibition and antioxidant defense.
We thank the Reviewer for this suggestion, we have improved this aspect in the new version of the manuscript on page 14, lines 428-431, as follows:
“…our findings suggest that novel targeted therapies for FXD patients might be focused on antioxidant defense and pharmacological inhibition of the PTP, as for example the compound TR001, which inhibits the PTP independently of CyPD and prevents thiol oxidation [51], or other PTP specific inhibitors [52,53].”

Reviewer 2 Report

Comments and Suggestions for Authors

Reviewer comments and suggestions

The authors in this study investigated the mitochondrial mechanisms that are involved in the Fragile X-related disorders (FXD) patients and examined mitochondrial morphology and bioenergetics in fibroblasts derived from patients. 

Analysis of mitochondrial oxidative phosphorylation in situ revealed lower respiration in premutation fibroblasts. This study explained the mitochondrial mechanisms which are involved in the FXD phenotypes, and indicated altered mitochondrial function and morphology. Overall, the authors suggested that mitochondria are novel drug targets to relieve the FXD symptoms

Overall, the manuscript was well written. However, a few major concerns or comments needed to be explained or modified.

  1. Line 49-50 Please explain the cited references rather than citing only
  2. Line 56 Please include the data on which neurodegenerative diseases
  3. Line 72-73 The authors need to cite references for this
  4. Line 77 The authors could comprehensively discuss this diseases, but I did not find it in the introduction part
  5. Comments for Figure 1 The significant results need to be illustrated by which group they compared, I mean, they can use 

indicators for this

  1. Line 277-278 Please explain the result of the pathogenesis they are discussing here
  2. I would be nice if the authors mentioned a table or figure in the text of the discussion section so that readers could easily understand the point they made.
  3. Line 326-327 is there any possible reason for this observation
  4. Comments for reference number I think there was no square bracket, please check the MDPI guidelines and please proofread the reference numbers 6, 33.

Author Response

Dear Editor,
We would like to greatly thank you and the Reviewers for the opportunity to improve our work, which in our opinion has been ameliorated by the revision process.
We have addressed all the Reviewer’s points as you can find in detail in the point by point-response to Reviewers below.
Please find attached our revised manuscript and figures.
Sincerely,
Valentina Giorgio

Reviewer 2
The authors in this study investigated the mitochondrial mechanisms that are involved in the Fragile X-related disorders (FXD) patients and examined mitochondrial morphology and bioenergetics in fibroblasts derived from patients.
Analysis of mitochondrial oxidative phosphorylation in situ revealed lower respiration in premutation fibroblasts. This study explained the mitochondrial mechanisms which are involved in the FXD phenotypes, and indicated altered mitochondrial function and morphology. Overall, the authors suggested that mitochondria are novel drug targets to relieve the FXD symptoms
Overall, the manuscript was well written. However, a few major concerns or comments needed to be explained or modified.
1. Line 49-50 Please explain the cited references rather than citing only
We thank the Reviewer for this comment, we have added a small paragraph explaining the mechanisms required on page 2, lines 50-54, as follows:
“FMRP functions as an mRNA-binding translation suppressor, but recent findings have enormously expanded its proposed roles to the involvement in RNA-, channel- and protein-binding leading to calcium signaling, activity-dependent critical period development and excitation-inhibition neural circuitry balance.”
2. Line 56 Please include the data on which neurodegenerative diseases
The text has been modified on page 2, lines 65-67, as follows:
“Pharmacological treatments causing PTP desensitization or targeting the CyPD were shown to be effective against neurodegeneration in autoimmune encephalomyelitis, superoxide dismutase 1- associated or Alzheimer’s disease models [10] (Scheme 1).”
3. Line 72-73 The authors need to cite references for this
We thank this Reviewer for the comment, the appropriate reference has been added at line 82, page 3.
4. Line 77 The authors could comprehensively discuss this diseases, but I did not find it in the introduction part
The abbreviations and discussion for the indicated diseases are in the Introduction, on page 1, lines 37-47.
5. Comments for Figure 1 The significant results need to be illustrated by which group they compared, I mean, they can use indicators for this
We would like to thank the Reviewer. We have indicated the groups that are significantly different with comparing indicators, in the revised Figure 1.
6. Line 277-278 Please explain the result of the pathogenesis they are discussing here
The text has been modified on page 10, lines 244-246, as follows:
“Other studies supported the role of mitochondrial dysfunction in the pathogenesis of FXTAS, indicating lower mitochondrial membrane potential, and decreased basal respiration and ATP synthesis or iron accumulation in PM patients [14,15,21,34,35].”
7. I would be nice if the authors mentioned a table or figure in the text of the discussion section so that readers could easily understand the point they made.
We would like to thank the Reviewer for this suggestion. The revised version of the Discussion includes accordingly the new Scheme 2, on page 10.
8. Line 326-327 is there any possible reason for this observation
We thank the Reviewer for the comment, our interpretation of the results is presented in the manuscript just below the aforementioned observation on page 11, lines 295-300, as follows:
“Finally, we showed that the treatment with arachidonic acid but not with Bz 423 increased apoptosis in FXD fibroblasts. Bz 423 is the PTP-inducer which was demonstrated to activate the channel by direct binding to the OSCP subunit of ATP synthase, and to displace both CyPD and IF1 from their binding sites [22,28]. This compound was not efficient in inducing cell apoptosis probably due to the high levels of CyPD and IF1 masking its binding site in FXD fibroblasts [28].”
9. Comments for reference number I think there was no square bracket, please check the MDPI guidelines and please proofread the reference numbers 6, 33.
Thank you we have checked the MDPI guidelines.
